# Assessment of the Effectiveness of Vibration Therapy and Passive Rest on the Recovery of Muscular Strength and Plasma Lactate Levels in the Upper Limbs after Intense Anaerobic Exercise in Elite Boxers and Kickboxers

Wiesław Chwała [1], Wacław Mirek [2], Tadeusz Ambroży [2], Wojciech Wąsacz [2], Klaudia Jakubowska [3] and Łukasz Rydzik [2,*]

[1]  Department of Biomechanics, Faculty of Physical Education and Sport, University of Physical Education, 31-571 Kraków, Poland; wieslaw.chwala@awf.krakow.pl

[2]  Institute of Sports Sciences, Faculty of Physical Education and Sport, University of Physical Education, 31-571 Kraków, Poland; waclaw.mirek@awf.krakow.pl (W.M.); tadek@ambrozy.pl (T.A.); wojciech.wasacz@doctoral.awf.krakow.pl (W.W.)

[3]  Department of Nursing, Faculty of Health Sciences, Vincent Pol University, 20-816 Lublin, Poland; klau.jakubowska@gmail.com

*  Correspondence: lukasz.rydzik@awf.krakow.pl; Tel.: +48-730-696-377

**Abstract:** Background: High-intensity anaerobic physical training frequently leads to muscle fatigue among boxers and kickboxers. Vibrational therapy (VT) and passive rest (PR) have been employed as methods to enhance muscular recovery and performance. This study evaluates the effectiveness of these two recovery methods on upper limb muscle strength and lactate levels in plasma after high-intensity exertion. Methods: Eighteen elite boxers and kickboxers, aged 19–32 years, underwent tests employing VT and PR as recovery methods in a controlled, crossover study. Muscle performance was assessed via isokinetic dynamometry, and lactate levels were measured pre-exercise, post-exercise, and post-recovery. The study adhered to the Declaration of Helsinki guidelines and was approved by the relevant bioethics committee. Results: The results showed that VT led to a faster recruitment of muscle fibers and improved muscle endurance as indicated by decreased fatigue work indices compared to PR. However, no significant differences were observed in peak torque or lactate levels between the two recovery methods. The VT group exhibited quicker recovery times in torque generation and better performance in fatigue resistance. Conclusions: VT appears to provide superior muscular recovery compared to PR following intense anaerobic effort, particularly in terms of muscle strength endurance and activation speed. These findings support the potential of VT in sports recovery protocols, although similar lactate response suggests that metabolic recovery rates are not significantly affected.

**Keywords:** vibration therapy; passive rest; muscle recovery; lactate levels; isokinetic dynamometry

## 1. Introduction

Physical training in boxers and kickboxers often results in muscle fatigue causing soreness and swelling, which leads to post-traumatic scarring and consequently reduces their performance and muscle strength. This condition is frequently a result of inadequate rest and biological regeneration necessary for proper muscle preparation for the next training session [1]. When recovery from the damage caused by physical activity is insufficient, the risk of injury increases, and difficulties arise in performing subsequent physical tasks [2], creating a barrier to continued physical engagement [3–5].

A critical component of understanding exercise-induced muscle fatigue and recovery involves the measurement of plasma lactate levels. Plasma lactate is a key indicator of anaerobic metabolism and muscle fatigue. During high-intensity anaerobic exercise, lactate

is produced faster than it can be removed, leading to an accumulation in the blood [6,7]. Elevated lactate levels are associated with muscle acidosis, which contributes to the sensation of fatigue and can impair muscle performance [4,8]. Monitoring lactate levels provides valuable insights into the metabolic demands placed on athletes and the effectiveness of different recovery strategies [9]. Achieving post-exercise muscle relaxation associated with the removal of harmful by-products of energy metabolism and the restoration of motor potential is a highly desirable and essential element of post-exercise recovery [6].

Currently, various methods are used in high-level sports to regenerate tired and damaged muscles during exercise. Properly selected biological regeneration shortens the individual phases and accelerates the restitution process, allowing for greater exercise capacity due to the hypercompensation phase of the body's motor potential [8,10].

Vibrations, as a factor inducing strong mechanical stimuli in the neuromuscular system, skeletal tissues, and muscles, have been studied in medicine [3]. It is now known that whole-body vibration (WBV) is used in sports training to increase muscle strength, as demonstrated by research from various scientists [11–13]. Authors [14–18] have also shown in their publications that vibrations are effective in alleviating muscle pain caused by physical exertion [19]. Unfortunately, current studies provide limited information on how the reduction of muscle pain, especially from short-term intense physical exertion, correlates with the recovery and potential hypercompensation of muscle strength. The use of vibrations as an innovative post-exercise recovery method is becoming an important topic that is gaining increasing popularity [20].

Few authors who have applied vibration therapy after physical exercise report positive effects on muscle recovery, although these studies did not involve elite or professional athletes [21]. On the other hand, there is information suggesting that using vibrations within 24 h post-exercise may be detrimental to muscle strength and biological regeneration during this period [22,23]. During vibration therapy, small muscle contractions occur, involving muscle lengthening and shortening. It is possible that the additional work required by the muscles during WBV may prolong the damage to the muscles, consequently increasing the loss of muscle strength [24]. This theory is supported by studies suggesting that increased muscle contractions lead to decreased muscle strength [25]. Other authors report that WBV does not aid muscle recovery post-exercise, as muscle fatigue changes the effectiveness of WBV as a recovery tool [26,27]. They suggest that the excitation–contraction coupling is impaired after muscle fatigue, reducing calcium release, leading to an inability to activate force-generating muscle fiber structures. It has also been suggested that muscle injury from physical exertion reduces the sensitivity to stretch reflex and muscle stiffness, decreasing the mechanisms that enhance muscle strength [28].

Given these mixed findings, our study aimed to assess and compare the impact of an original vibration therapy protocol and passive rest on the recovery of upper limb muscle strength and plasma lactate levels after intense anaerobic exercise in elite boxers and kickboxers. Compensation of upper limb muscle strength was assessed using isokinetic dynamometry, functional performance tests, and lactate measurement.

The aim of this experiment was to assess and compare the impact of a specifically designed vibration therapy protocol, which we refer to as 'original vibration therapy', and passive rest on the compensation of upper limb muscle strength and plasma lactate levels after intense anaerobic exercise, predominantly anaerobic exercise. This original vibration therapy involves a unique combination of frequency, amplitude, and duration tailored for optimal muscle recovery. This study includes both boxers and kickboxers to capture the effects of recovery methods on athletes with varying muscle usage patterns. As boxers primarily utilize upper limb strength, while kickboxers engage both upper and lower limbs, this approach provides a comprehensive understanding of the recovery process across different combat sports disciplines.

## 2. Materials and Methods

### 2.1. Study Design

The study was conducted according to the guidelines of the Declaration of Helsinki and approved by the Bioethics Committee of the Regional Medical Chamber (No. 287/KBL/OIL/2020, approved on 30 December 2020). This controlled crossover study aimed to compare the effectiveness of vibration therapy (VT) and passive rest (PR) on the recovery of muscular strength and plasma lactate levels in elite boxers and kickboxers following predominantly anaerobic exercise. The exercise protocol was designed to replicate the intensity and type of activity typical in their training sessions.

### 2.2. Participants

Although 18 elite boxers and kickboxers aged 19–32 years were initially recruited, only 14 completed all phases of the study due to personal reasons or scheduling conflicts. All participants were free from upper limb injuries within 6 months prior to the study and had fully healed from any other injuries that could affect the results.

Participants were recruited using purposive sampling from local sports clubs and training centers known for producing high-level athletes. Purposive sampling was chosen to ensure that all participants had the requisite high level of training and performance needed for this study. Invitations were extended to 25 potential participants who met the inclusion criteria, which included having at least a master class ranking and a minimum of 8 years of training experience. This specific number was chosen based on the availability and willingness of eligible athletes to participate in a rigorous and controlled experimental protocol. Out of the 25 invited, 7 declined to participate due to scheduling conflicts or personal reasons, resulting in a final sample size of 18 individuals.

The sample size was calculated using the G*Power software (version: 3.1.9.7; Dusseldorf University, Dusseldorf, Germany). The parameters used for this calculation were based on an effect size (Cohen's d) estimated from previous similar studies examining the effects of vibration therapy on muscle recovery, an alpha level ($\alpha$) of 0.05, and a statistical power ($1 - \beta$) of 0.80. Specifically, we assumed an effect size of 0.85, which is considered large according to Cohen's conventions. This calculation indicated that a minimum of 18 participants would be sufficient to detect significant differences with the desired power and significance level. The study was conducted in a controlled laboratory setting at the University of Physical Education, ensuring consistency in testing conditions.

The 18 participants engaged in the study twice, serving as both the experimental and control groups, with group assignment determined by the type of biological regeneration applied (vibration therapy—experimental group, passive rest—control group). Fourteen athletes completed all the required trials. The characteristics of the study group are presented in Table 1.

**Table 1.** Somatic characteristics of the participants.

|  | Average | Min | Max | SD |
|---|---|---|---|---|
| Body Mass (BM) [kg] | 76.0 | 52 | 90 | 9.67 |
| Body Height (BH) [cm] | 178.3 | 165 | 189 | 7.77 |
| Body Mass Index (BMI) [kg/m$^2$] | 23.9 | 18 | 28 | 2.66 |
| Age [years] | 23.0 | 19 | 32 | 5.81 |
| Fat [%] | 10.4 | 5 | 21 | 3.73 |
| Total Body Water (TBW) [%] | 62.9 | 56 | 67 | 3.04 |

SD—standard deviation, Min—minimum value, Max—maximum value.

### 2.3. Inclusion and Exclusion Criteria

Inclusion criteria were as follows:

- Minimum of 8 years of training experience
- No upper limb injuries within the past 6 months
- Active training phase during the preparatory period
- Up-to-date medical examinations

Exclusion criteria included:

- Injuries affecting upper limb functionality within the past 6 months
- Lack of current medical exams
- Lack of consent to participate

### 2.4. Biological Regeneration and Allocation

Recovery strategies are essential in enhancing muscle repair and function following intense physical activity. In this study, we compared two recovery methods: vibration therapy (VT) and passive rest (PR). Participants were randomly assigned to begin with either VT or PR using a random number generator to ensure unbiased allocation.

Vibration Therapy (VT): This involved a 15 min session of vibration massage targeting the upper limb muscles. The therapy was administered using a Vitberg vibration therapy device (Rehabilitation Medical Equipment VITBERG, Nowy Sącz, Poland) set to frequencies ranging from 20 to 50 Hz. The vibration therapy was applied to the following muscle groups:

- Biceps Brachii: The therapy was applied for 5 min per arm.
- Triceps Brachii: The therapy was applied for 5 min per arm.
- Deltoid Muscles: The therapy was applied for 5 min per arm.

Each muscle group received targeted vibrations aimed at enhancing blood flow and promoting muscle relaxation. The sessions were divided evenly to ensure comprehensive coverage of the primary muscles involved in boxing and kickboxing, especially the extensors and flexors of the elbow joints, which perform essential work when performing straight punches.

Passive Rest (PR): Participants in this group were instructed to rest passively in a supine position for 15 min without any additional interventions. This served as a control to compare the effects of VT.

Crossover Design: After a washout period to eliminate any residual effects of the first treatment, participants switched to the alternate recovery method. This crossover design allowed each participant to serve as their own control, thereby reducing inter-individual variability. A washout period of 7 days was implemented between the two experimental sessions to eliminate any residual effects from the first intervention.

Outcome Measures: Muscle performance was assessed using isokinetic dynamometry to measure peak torque and endurance. Plasma lactate levels were measured at three time points: after the warm-up, immediately post-exercise, and post-recovery. These measures provided insights into both the physical and metabolic aspects of muscle recovery.

The 18 participants engaged in the study twice, serving as both the experimental and control groups, with group assignment determined by the type of recovery method applied. Fourteen athletes completed all the required trials.

### 2.5. Blinding and Allocation Concealment

Due to the nature of the interventions (VT vs. PR), blinding participants was not possible as they could clearly distinguish between the two recovery methods. However, the assessors who measured outcomes were blinded to the group assignments to minimize bias. Allocation concealment was maintained by having an independent researcher generate the random allocation sequence and assign participants to the groups.

### 2.6. Study Procedure

The study was conducted in the morning and divided into several stages:

#### 2.6.1. Stage I: Preliminary Measurements

Participants' body height and weight were measured using a Tanita BC-418 MA body composition analyzer (Tanita Corporation, Tokyo, Japan) with an accuracy of 0.01 m and 0.01 kg, respectively. Body fat percentage was also measured using the same device. Resting plasma lactate levels were determined through blood samples analyzed using a Lactate Plus meter (Nova Biomedical, Waltham, MA, USA).

#### 2.6.2. Stage II: Warm-Up

A 15 min individual standard warm-up for boxers and kickboxers was performed. This included dynamic stretching and light aerobic exercises to prepare the muscles for subsequent testing.

#### 2.6.3. Stage III: Baseline Strength Assessment

Maximum punching force was measured based on three punches with each upper limb using accelerometric sensors (BioPac Systems Inc., Goleta, CA, USA) attached to a boxing trainer and the participant's boxing gloves. The highest force recorded for each limb served as the baseline for determining minimum punching force levels in the main experiment. While these sensors provide accurate measurements of acceleration, it is essential to differentiate this from force production capabilities. The data suggest potential force but are not direct measurements of muscular force. Muscle force was estimated indirectly from body mass, movement speed, and time.

#### 2.6.4. Stage IV: Isokinetic Strength Testing

Maximum torque capabilities were measured under isokinetic contraction conditions for the flexors and extensors of the elbow joints at an angular velocity of $\omega = 300°\text{s}^{-1}$ using a Biodex System 4 Pro dynamometer (Biodex Medical Systems, Shirley, NY, USA). This angular velocity was chosen based on literature suggesting that it closely mimics the rapid, explosive movements common in boxing and kickboxing, providing a relevant assessment of muscle function under sport-specific conditions. Participants performed three maximal contractions for each muscle group, and the highest value was recorded. In this study, the 'work' performed by the muscles during each test was calculated by integrating the force–displacement curve across the range of motion during isokinetic testing. This calculation provides a quantitative assessment of the total mechanical work output by the muscles, reflecting the physical effort exerted throughout the exercise bout. This method ensures a comprehensive analysis of muscular endurance and the effectiveness of the recovery interventions.

#### 2.6.5. Stage V: Muscle Fatigue Induction

Participants performed auxotonic exercises over three rounds of 180 alternating straight punches in each round using a boxing trainer over 120 s at no less than 80% of their maximum punching force. Each round was separated by a 1 min passive rest. Physical exertion and muscle fatigue were monitored using the accelerometric sensors to ensure that participants reached a state of fatigue. Punch force was continuously measured using accelerometric sensors to ensure consistency in exertion levels across all participants. Muscle fatigue was monitored by analyzing the deceleration in punch force over time, as recorded by the accelerometric sensors.

#### 2.6.6. Stage VI: Recovery Interventions

Participants were randomly assigned to either the VT or PR group. VT consisted of a 15 min vibration massage of the upper limb muscles using a Vitberg vibration therapy device with frequencies ranging from 20 to 50 Hz. PR involved a 15 min passive rest in a

supine position. Isokinetic tests were repeated immediately after the recovery period. All participants underwent a familiarization session with the Biodex dynamometer at 300°/s to ensure accurate and consistent measurements during the testing phase.

### 2.7. Plasma Lactate Measurement

Plasma lactate levels were measured three times during each session: after warm-up, immediately post-exercise, and post-recovery. Blood samples for lactate measurement were taken from the fingertip and measured with a Lactate Scout device (EKF Diagnostics, Barleben, Germany).

### 2.8. Statistical Analysis

Data were analyzed using Statistica 13.3 software. Basic descriptive statistics (mean, standard deviation, minimum, and maximum) were calculated. The normality of all measured datasets was assessed using the Shapiro–Wilk test, and only normally distributed data were analyzed using parametric tests. Homogeneity of variances was evaluated with the Levene test. Repeated measures analysis of variance (ANOVA) was employed to identify significant differences between groups, with Tukey's post hoc test used for pairwise comparisons. A *p*-value of <0.05 was considered statistically significant.

## 3. Results

### 3.1. Peak Torque Dynamics (PTQ)

The application of vibration therapy (VT) resulted in improved dynamics of peak torque generation. The VT group achieved peak torque faster than the passive rest (PR) group, indicating quicker recovery of muscle contraction ability (Table 2; Appendix A— Figure A1). The mean time to reach peak torque (T_PTQ) was significantly shorter in the VT group, suggesting better muscle performance and faster response to training stimuli.

**Table 2.** Descriptive statistics of the analyzed variables.

| Variables | Average | | *p* | Min | | Max | | SD | |
|---|---|---|---|---|---|---|---|---|---|
| | LBV | PR | | LBV | PR | LBV | PR | LBV | PR |
| Peak Torque Right Extensors (PTQ REXT) [Nm] | 56 | 58 | 0.26019 | 32 | 34 | 85 | 79 | 11.1 | 11.0 |
| Peak Torque Left Extensors (PTQ LEXT) [Nm] | 58 | 60 | 0.63567 | 29 | 30 | 88 | 88 | 14.9 | 14.7 |
| Peak Torque Right Flexors (PTQ RFLEX) [Nm] | 47 | 45 | 0.18479 | 31 | 29 | 70 | 73 | 9.0 | 10.0 |
| Peak Torque Left Flexors (PTQ LFLEX) [Nm] | 45 | 46 | 0.85424 | 28 | 30 | 64 | 69 | 9.0 | 10.1 |
| Peak Torque to Body Weight Right Extensors (PTQ_BW REXT) [%] | 74 | 77 | 0.05253 | 50 | 59 | 95 | 96 | 9.9 | 9.4 |
| Peak Torque to Body Weight Left Extensors (PTQ_BW LEXT) [%] | 76 | 79 | 0.37649 | 53 | 51 | 100 | 109 | 13.2 | 13.8 |
| Peak Torque to Body Weight Right Flexors (PTQ_BW RFLEX) [%] | 63 | 61 | 0.17405 | 43 | 39 | 105 | 90 | 12.3 | 12.4 |
| Peak Torque to Body Weight Left Flexors (PTQ_BW LFLEX) [%] | 59 | 61 | 0.76688 | 41 | 38 | 86 | 91 | 10.4 | 11.6 |
| Time to Peak Torque Right Extensors (T_PTQ REXT) [ms] | 331 | 363 | 0.11361 | 10 | 10 | 440 | 460 | 106.1 | 91.6 |
| Time to Peak Torque Left Extensors (T_PTQ LEXT) [ms] | 353 | 347 | 0.53673 | 10 | 160 | 790 | 440 | 116.8 | 66.1 |
| Time to Peak Torque Right Flexors (T_PTQ RFLEX) [ms] | 347 | 405 | 0.12209 | 60 | 190 | 610 | 660 | 137.3 | 126.0 |
| Time to Peak Torque Left Flexors (T_PTQ LFLEX) [ms] | 364 | 444 | **0.00759** | 50 | 60 | 650 | 760 | 139.0 | 133.3 |
| Work to Body Weight Right Extensors (WRK_BW REXT) [J] | 88 | 96 | 0.10068 | 61 | 66 | 118 | 127 | 14.0 | 16.5 |
| Work to Body Weight Left Extensors (WRK_BW LEXT) [J] | 90 | 96 | 0.23203 | 62 | 59 | 122 | 130 | 17.5 | 17.9 |
| Work to Body Weight Right Flexors (WRK_BW RFLEX) [J] | 85 | 89 | 0.92173 | 60 | 37 | 119 | 135 | 18.4 | 23.2 |
| Work to Body Weight Left Flexors (WRK_BW LFLEX) [J] | 87 | 89 | 0.97878 | 57 | 41 | 117 | 170 | 18.9 | 24.7 |
| Total Work Right Extensors (TOT_WVRK REXT) [J] | 360 | 402 | **0.01711** | 214 | 194 | 535 | 578 | 78.4 | 92.0 |
| Total Work Left Extensors (TOT_WRK LEXT) [J] | 372 | 401 | 0.36831 | 205 | 234 | 621 | 637 | 107.6 | 103.7 |
| Total Work Right Flexors (TOT_WRK RFLEX) [J] | 364 | 381 | 0.96800 | 197 | 144 | 553 | 615 | 100.5 | 115.0 |
| Total Work Left Flexors (TOT_WRK LFLEX) [J] | 374 | 377 | 0.70304 | 199 | 168 | 598 | 596 | 107.0 | 112.5 |

**Table 2.** *Cont.*

| Variables | Average | | *p* | Min | | Max | | SD | |
|---|---|---|---|---|---|---|---|---|---|
| | LBV | PR | | LBV | PR | LBV | PR | LBV | PR |
| Average Power Right Extensors (AVG_POW REXT) [W] | 103 | 110 | 0.11749 | 63 | 62 | 145 | 154 | 20.7 | 21.7 |
| Average Power Left Extensors (AVG_POW LEXT) [W] | 104 | 111 | 0.41284 | 54 | 68 | 151 | 165 | 26.4 | 25.4 |
| Average Power Right Flexors (AVG_POW RFLEX) [W] | 103 | 99 | 0.23974 | 58 | 34 | 152 | 157 | 24.8 | 29.6 |
| Average Power Left Flexors (AVG_POW LFLEX) [W] | 103 | 100 | 0.29445 | 53 | 47 | 157 | 156 | 27.0 | 28.4 |
| Average Peak Torque Right Extensors (AVG_PTQ REXT) [Nm] | 51 | 54 | 0.17709 | 29 | 31 | 80 | 76 | 11.6 | 11.2 |
| Average Peak Torque Left Extensors (AVG_PTQ LEXT) [Nm] | 52 | 55 | 0.38166 | 27 | 28 | 84 | 84 | 14.9 | 14.3 |
| Average Peak Torque Right Flexors (AVG_PTQ RFLEX) [Nm] | 44 | 43 | 0.32650 | 29 | 26 | 65 | 70 | 8.5 | 9.5 |
| Average Peak Torque Left Flexors (AVG_PTQ LFLEX) [Nm] | 42 | 43 | 0.74334 | 25 | 28 | 63 | 64 | 9.2 | 9.3 |
| Acceleration Time Right Extensors (T_ACC REXT) [ms] | 110 | 116 | 0.36894 | 10 | 80 | 160 | 170 | 25.8 | 21.0 |
| Acceleration Time Left Extensors (T_ACC LEXT) [ms] | 126 | 112 | 0.99721 | 80 | 80 | 660 | 150 | 91.2 | 17.5 |
| Acceleration Time Right Flexors (T_ACC RFLEX) [ms] | 172 | 197 | **0.00947** | 120 | 120 | 250 | 360 | 36.4 | 56.5 |
| Acceleration Time Left Flexors (T_ACC LFLEX) [ms] | 173 | 195 | **0.04203** | 120 | 120 | 330 | 570 | 40.4 | 72.8 |
| Deceleration Time Right Extensors (T_DEC REXT) [ms] | 238 | 233 | 0.89468 | 140 | 180 | 430 | 350 | 62.7 | 37.8 |
| Deceleration Time Left Extensors (T_DEC LEXT) [ms] | 246 | 232 | 0.57064 | 180 | 180 | 460 | 370 | 59.0 | 41.8 |
| Deceleration Time Right Flexors (T_DEC RFLEX) [ms] | 156 | 158 | 0.90776 | 90 | 90 | 260 | 240 | 42.1 | 36.6 |
| Deceleration Time Left Flexors (T_DEC LFLEX) [ms] | 160 | 169 | 0.38700 | 100 | 90 | 290 | 260 | 39.6 | 39.3 |
| Agonist to Antagonist Ratio Right (AGN_ANT RAT RIGHT) [%] | 86 | 80 | **0.01835** | 57 | 56 | 136 | 117 | 17.2 | 17.9 |
| Agonist to Antagonist Ratio Left (AGN_ANT RAT LEFT) [%] | 80 | 80 | 0.54263 | 57 | 49 | 137 | 117 | 17.6 | 15.1 |
| Work Fatigue Right Extensors (LBV_WRK_FAT REXT) [J/J] | −4.5 | 5.4 | **0.0044** | −31.2 | −34.2 | 26.0 | 32.4 | 11 | 11 |
| Work Fatigue Left Extensors (LBV_WRK_FAT LEXT) [J/J] | −3.8 | 5.6 | **0.0012** | −37.8 | −7.4 | 42.2 | 13.8 | 16 | 6 |
| Work Fatigue Right Flexors (LBV_WRK_FAT RFLEX) [J/J] | 1.8 | 4.8 | 0.13268 | −26.8 | −13.9 | 20.8 | 24.2 | 10 | 7 |
| Work Fatigue Left Flexors (LBV_WRK_FAT LFLEX) [J/J] | 5.3 | 5.1 | 0.24539 | −16.4 | −12.9 | 13.0 | 18.3 | 7 | 8 |
| Lactate (LA) [mmol] | 5.2 | 5.3 | 0.91672 | 0.5 | 0.5 | 14.8 | 12.5 | 4.5 | 4.2 |
| Relative SBFT Index [bmp/Nkg$^{-1}$] | 2.42 | 2.42 | 0.74882 | 2.02 | 1.99 | 3.66 | 3.49 | 0.44 | 0.42 |

LBV—vibration therapy group, PR—passive rest group, SD—standard deviation, Min—minimum value, Max—maximum value. *p*—level of significance of variation; statistically significant values are shown in bold.

### 3.2. Acceleration and Reaction Time (T_ACC)

The acceleration time (T_ACC), measuring the time taken for muscles to reach a set velocity, was significantly shorter in the VT group. This reflects a faster response and potentially more efficient recruitment of muscle fibers following vibration therapy, which could be crucial in sports requiring rapid muscle reactions (Table 2; Appendix A—Figure A2). It is important to clarify that while we measured acceleration, this should not be directly equated with muscle force. Acceleration indicates the potential for force but does not confirm it without further biomechanical assessments.

### 3.3. Muscle Work and Fatigue (WRK_FAT)

Muscle fatigue indices (WRK_FAT) indicated that the VT group experienced a lower rate of fatigue compared to the PR group. This suggests that VT contributes to better muscle endurance and maintains performance over time, mitigating the effects of fatigue during recovery phases. Work was determined by integrating the force–displacement curve over the range of motion during isokinetic testing (Table 2; Appendix A—Figure A3).

### 3.4. Muscle Strength Balance

The agonist-to-antagonist ratio (AGN_ANT_RAT) was more favorable in the VT group. The improved ratio suggests a more symmetrical recovery of muscle groups, potentially reducing injury risk by maintaining better joint stability and muscle health. The agonist-to-

antagonist ratio was calculated by comparing the peak torque of opposing muscle groups during isokinetic testing. It is crucial to acknowledge the significance of measuring the agonist–antagonist ratio at consistent joint angles to ensure the validity of these assessments. Discrepancies in joint angles during measurement can lead to misleading interpretations of muscle balance and function. This study has taken steps to standardize joint angles across all measurements to address this concern, thereby enhancing the reliability of the agonist–antagonist ratio as an indicator of muscle health and balance (Table 2; Appendix A—Figure A4).

### 3.5. Total Muscle Work (TOT_WRK)

Despite the observed benefits in other variables, the total work (TOT_WRK) performed by the muscles during the tests was slightly higher in the PR group. This may indicate that although VT improves certain aspects of muscle function, it does not necessarily translate into increased total work performed during the recovery phase (Table 2; Appendix A—Figure A5).

### 3.6. Physiological Responses—Lactate Levels (LA)

Lactate levels were measured to assess the metabolic response to both recovery methods. Although initial lactate levels post-exercise were comparable between groups, the VT group showed a more significant reduction in lactate levels after recovery, suggesting more efficient removal of metabolic by-products (Table 2).

The results of lactate level measurements in the blood indicated similar acidification levels. Lactate levels did not show statistically significant differences between the VT and PR groups at any of the measured time points obtained in both studies, in the LBV and PR groups. However, the serum lactate level differed significantly between individual trials (intra-group) in both studies at the level of $p < 0.001$. The lowest values in both groups were obtained in the initial measurement before exercise. Subsequently, the values increased significantly to above 10 mmol, then returned to minimally above 4 mmol after biological regeneration (Appendix A—Figure A6).

## 4. Discussion

This study aimed to evaluate the effectiveness of vibration therapy (VT) and passive rest (PR) on the recovery of upper limb muscle strength and plasma lactate levels in elite boxers and kickboxers following predominantly anaerobic exercise. The results indicated that VT led to faster recovery of muscle function and improved muscle endurance compared to PR. While we observed quicker times to peak torque and reduced fatigue indices in the VT group, it is important to note that this study did not directly measure the recruitment of muscle fibers. The observed improvements in performance metrics suggest enhanced neuromuscular function, but further research using electromyography (EMG) or other techniques would be necessary to specifically assess muscle fiber recruitment.

The agonist-to-antagonist ratio is an important measure in this study as it provides insights into muscle balance and joint stability. A more favorable ratio suggests symmetrical recovery of muscle groups, which is crucial for reducing injury risk and maintaining overall muscle health. Research indicates that maintaining a balanced agonist–antagonist ratio helps in optimizing athletic performance and minimizing injury risks.

The findings support the potential of VT in sports recovery protocols, although similar lactate responses suggest that metabolic recovery rates are not significantly affected. Further research is needed to confirm these findings and explore the long-term effects of VT on muscle recovery and performance.

The results of the studies conducted by Chapman, Barnes, Dabbs, and Chwała and Pogwizd indicate discrepancies in the presented views, allowing the conclusion that the authors did not always achieve improvements in the measured parameters after applying vibration therapy following exercise or during breaks between exercise sets [21,24,25,28]. This may be due to the fact that each study used different vibration frequencies and amplitudes. The studies also varied in terms of the duration and location of the vibration application, which undoubtedly could have significantly impacted the final effects of the

therapy. However, recent studies have shown positive observations regarding the use of vibration as a superior form of intervention over passive rest in biological regeneration [18].

Whole-body vibration (WBV) treatments are an innovative form of intervention in the training process, and to date, only a few studies have examined the effect of combining different frequencies and amplitudes of vibration. As indicated by research [29], beneficial effects can be achieved by smoothly changing the frequency and amplitude of vibration applied during a WBV program. Similar methodologies have been used only in separate treatment cycles, comparing the effects of different frequencies and amplitudes on the achieved results. In most studies, authors utilized a strictly defined vibration stimulus (frequency, amplitude, acceleration) and examined its effect on measured variables.

Therefore, it seems reasonable to apply a vibration stimulus with variable frequency, amplitude, and duration, which will more effectively impact all types of muscle fibers and sensory receptors.

An essential condition for comparing the effects of varied interventions on the speed and effectiveness of biological regeneration is ensuring that the muscles of the study participants are fatigued to the same extent, in this case, during predominantly anaerobic exercise. When comparing loads in the exercise test, it should be noted that the relative values of the Rel. SBFT index [30] did not differ significantly between the LBV and PR groups. This indicates a similar level of fatigue in the flexor and extensor muscles of the elbow joints in both groups.

The use of dedicated vibrating mats allowed for localized vibration intervention in a comfortable resting position for the muscles controlling the upper limb joints, which could be significant for the relaxation and regeneration of muscles fatigued by exercise and its superiority over passive rest.

This is confirmed by the significantly shorter time to achieve peak torque (PTQ) for elbow flexor muscles in the LBV group compared to the PR group ($p < 0.01$) (Table 1). Time to peak torque is a measure of the time from the start of a muscular contraction to the point of highest torque development (Peak TQ). This value is an indicator of the muscle's functional ability to produce torque quickly. This variable illustrates the neuromuscular readiness to rapidly develop torque [31] and achieve maximum contraction [32]. A short T_PTQ time indicates a high level of proprioception in the joint area being studied.

A similar interpretation applies to the variable acceleration time (T_ACC), measured from the start of the movement to the moment of reaching the set speed during isokinetic contraction. The T_ACC variable for both the right and left limb flexors was significantly shorter in the LBV group compared to the PR group, by 25 ms (RFLEX), $p < 0.01$ and 22 ms (LFLEX), $p < 0.05$ (Table 2). Lower acceleration time values may indicate better recruitment ability of muscle fibers in the studied skeletal muscles and may be associated with shorter time needed to generate torque [32–34].

In our studies, besides faster recruitment of motor units in the muscle group after vibration intervention (LBV) compared to passive rest (PR), more favorable indices of strength endurance and muscle resistance to fatigue were also noted. The work fatigue (WRK_FAT) index, calculated as the ratio of the work performed by the muscles in the first 30% range of motion of all contractions to the work performed during the last 30% range of motion in the entire trial, characterizes the state of muscle fatigue in subsequent repetitions of the isokinetic test. The index value for elbow extensor muscles was significantly higher in the PR group compared to the LBV group, with a significance level for contrasts for the right limb of $p < 0.001$ and for the left limb $p < 0.05$. Higher WRK_FAT values in the PR group indicate that the work performed in the first 30% of the contraction was significantly higher than in the last 30% range of motion, indicating faster muscle fatigue during this test. In the LBV group, the index values were negative, indicating greater strength endurance of muscles after recovery using vibration interventions compared to passive rest.

As Gillet et al. argue, the risk of injury may be related to the strength balance of antagonistic muscle groups that provide joint stability [35]. The variable AGN_ANT_RAT (agonist/antagonist ratio), calculated as the ratio of peak PTQ values (maximum torque

of agonist muscles divided by the peak torque of antagonist muscles in the joint), is used by researchers to assess muscle balance. This index is important for evaluating muscle function [36,37]. Researchers indicate that the value of this index depends on age, sex, and training level [38]. Comparing the results of our studies regarding this index in the LBV and PR groups, it should be noted that a significantly higher AGN_ANT_RAT index was recorded in the LBV group for the right upper limb, which for all subjects was the dominant limb ($p < 0.05$). The corresponding index for the left upper limb did not show significant differences.

Another variable that showed significant contrasts between the LBV and PR groups was the total work (TOT_WRK) performed by the muscles during all movements in the isokinetic test. The total work value recorded in the PR group for the right elbow extensor was significantly higher by approximately 42 W compared to the analogous variable in the LBV group ($p < 0.05$). As shown above, the elbow extensor muscles performed most of the work in the first 30% of isokinetic contractions, which proved decisive for higher TOT_WRK values. The left elbow extensors also performed more work on average by about 28 W, but the differences were not statistically significant ($p < 0.05$).

Other variables, including the maximum torque levels (PRQ) and their relative values (PRQ_BW), despite some differences between the mean values in both groups, did not show significant contrasts at $p < 0.05$. In summary, it can be concluded that the type of post-exercise muscle recovery used did not significantly affect strength capabilities, but significantly differentiated the speed of motor unit recruitment and strength endurance in the group with proprietary vibration intervention compared to passive rest.

### 4.1. Limitations

This study has several limitations. First, the small sample size may limit the generalizability of the findings. Second, the lack of blinding of participants due to the nature of the interventions could introduce bias. Additionally, the focus on elite athletes may not make the results applicable to recreational athletes or those at different training levels. In our study, we focused mainly on elbow flexors and extensors because these muscle groups are heavily involved during straight punches performed by the players in the experiment, and they are characterized by a significant change in muscle length. In addition to the muscles of the shoulder joints, they also play a key role in performance in these sports and are easier to assess with the available equipment and time constraints. Future studies should aim to include a comprehensive assessment of shoulder muscles to provide a more complete understanding of the recovery processes in these athletes. Incorporating shoulder muscle assessments would allow for a more holistic evaluation of muscle recovery and performance, addressing this significant aspect of the athletic performance in boxing and kickboxing. Although we initially recruited 18 participants, only 14 completed all phases of the study. Four participants were unable to complete the study due to personal reasons or scheduling conflicts that arose during the course of the experiment. As a result, the final analysis was conducted with the data from these 14 participants. While this reduced sample size may impact the statistical power of our findings, the results still provide valuable insights into the effects of vibration therapy and passive rest on muscle recovery

### 4.2. Strengths

Despite these limitations, the study has notable strengths. The crossover design allows each participant to serve as their own control, reducing inter-individual variability. The use of objective measures such as isokinetic dynamometry and plasma lactate levels enhances the reliability of the results. The study also provides valuable insights into the application of VT in a practical sports setting.

### 4.3. Recommendations

Future studies should consider larger sample sizes and include recreational athletes to improve generalizability. Additionally, exploring other recovery modalities and their

effects on muscle performance and lactate clearance could provide a more comprehensive understanding of post-exercise recovery strategies. Including shoulder joint measurements would also provide a more complete assessment of muscle recovery in boxers.

## 5. Conclusions

1. The use of post-exercise vibration therapy with variable amplitude and frequency had a more favorable effect on the recovery of elbow joint muscles in the VT group compared to passive rest (PR).
2. Vibration therapy applied after predominantly anaerobic exercise positively affected the time to peak torque, acceleration of muscle function recovery, and endurance.
3. No significant differences were noted in the level of maximum and relative torque between the two recovery methods.
4. Lactate levels (LA) in both studies were similar at various time points of the experiments but significantly differed between trials in both groups. After anaerobic exercise, LA significantly increased from baseline values and then significantly decreased following recovery.

**Author Contributions:** Conceptualization, W.C. and T.A.; methodology, W.C., W.M. and K.J.; software, W.C.; validation, W.C., W.M. and W.W.; formal analysis, W.C.; investigation, W.C.; resources, W.C., W.M. and W.W.; data curation, W.C. and Ł.R.; writing—original draft preparation, W.C., T.A., W.W. and Ł.R.; writing—review and editing, W.C., T.A., W.W., Ł.R. and K.J.; visualization, W.C.; supervision, W.C., T.A. and Ł.R.; project administration, W.C., W.M., T.A., W.W. and Ł.R.; funding acquisition, W.C., T.A. and Ł.R. All authors have read and agreed to the published version of the manuscript.

**Funding:** This project is funded under the program of the Ministry of Science and Higher Education (Poland) 'Regional Excellence Initiative' for the years 2019–2022, project number 022/RID/2018/19, with a budget of 11,919,908 PLN; grant number 40/PB/RID/2022 with funding amount 56,500 PLN.

**Institutional Review Board Statement:** The study was conducted in accordance with the Declaration of Helsinki and approved by the Bioethics Committee of the Regional Medical Chamber (No. 287/KBL/OIL/2020), 30 December 2020.

**Informed Consent Statement:** Informed consent was obtained from all subjects involved in the study.

**Data Availability Statement:** The data presented in this study are available upon request from the corresponding author.

**Conflicts of Interest:** The authors declare no conflicts of interest.

## Appendix A

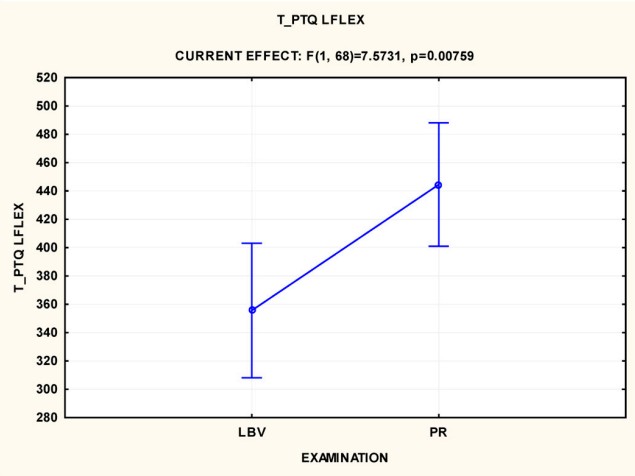

**Figure A1.** Time to reach peak PTQ in the LBV and PR groups (ms).

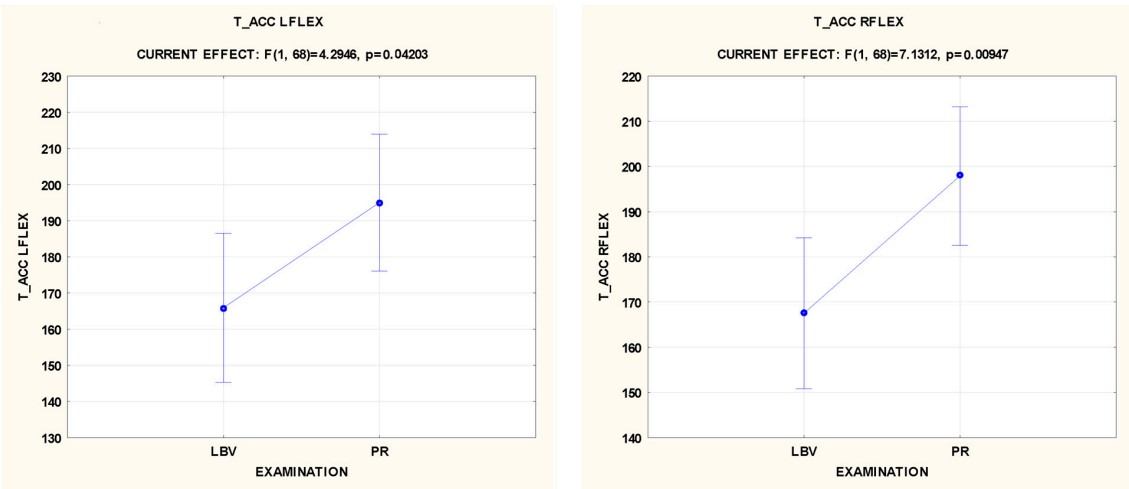

**Figure A2.** Acceleration time T_ACC in the LBV and PR groups (ms).

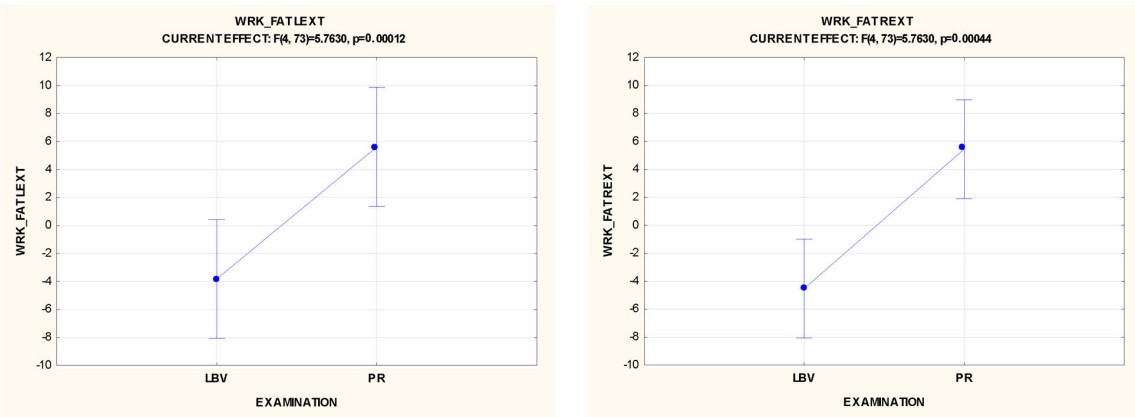

**Figure A3.** Work fatigue WRK_FAT in the LBV and PR groups (J).

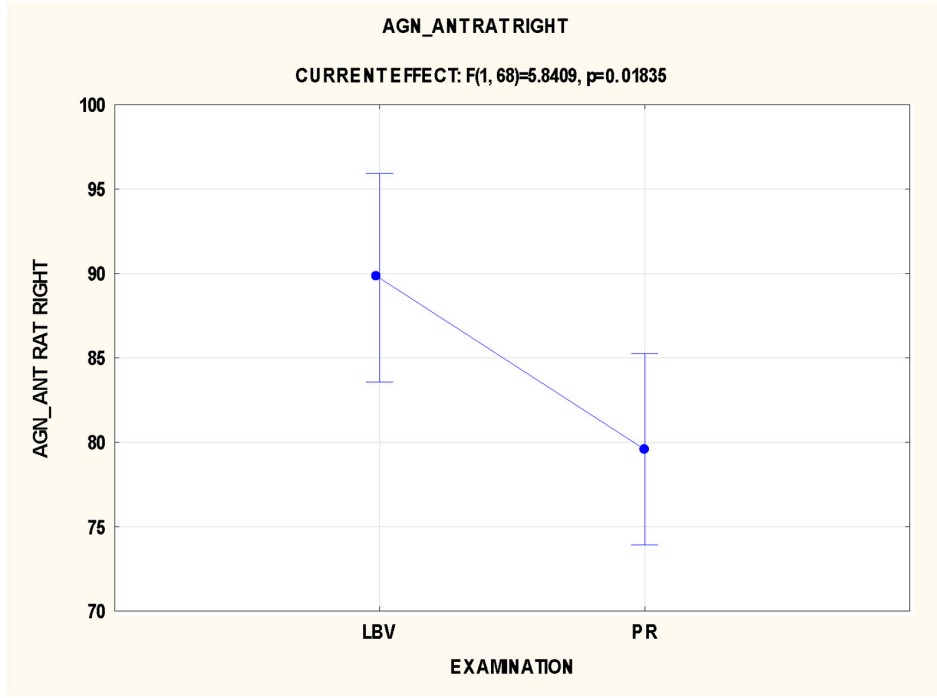

**Figure A4.** AGN_ANT_RAT ratio in the LBV and PR groups (Vat).

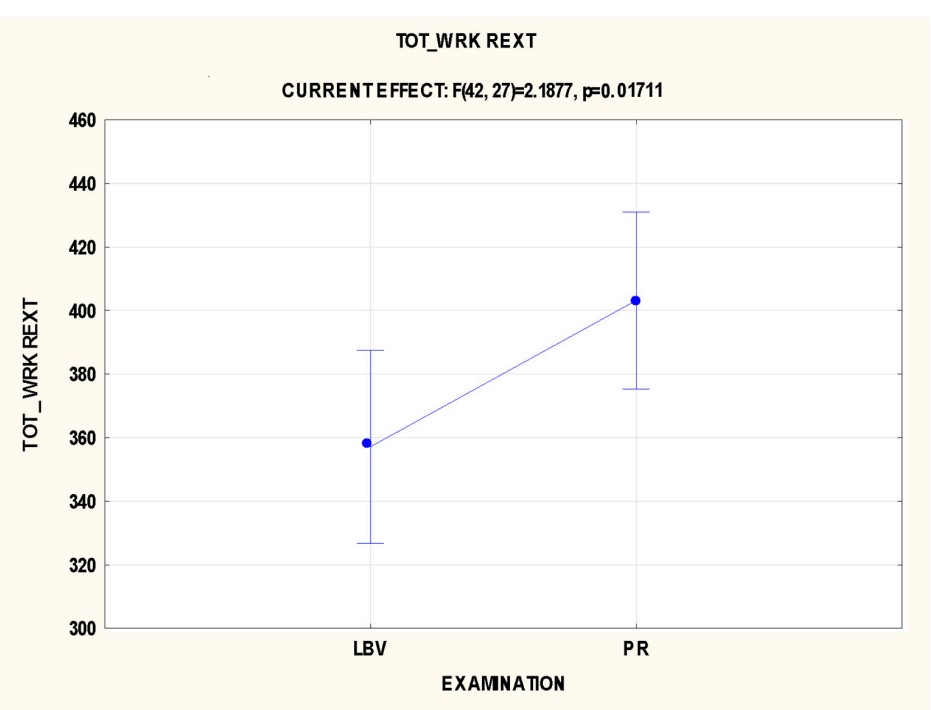

**Figure A5.** Total work TOT_WRK in the LBV and PR groups (J).

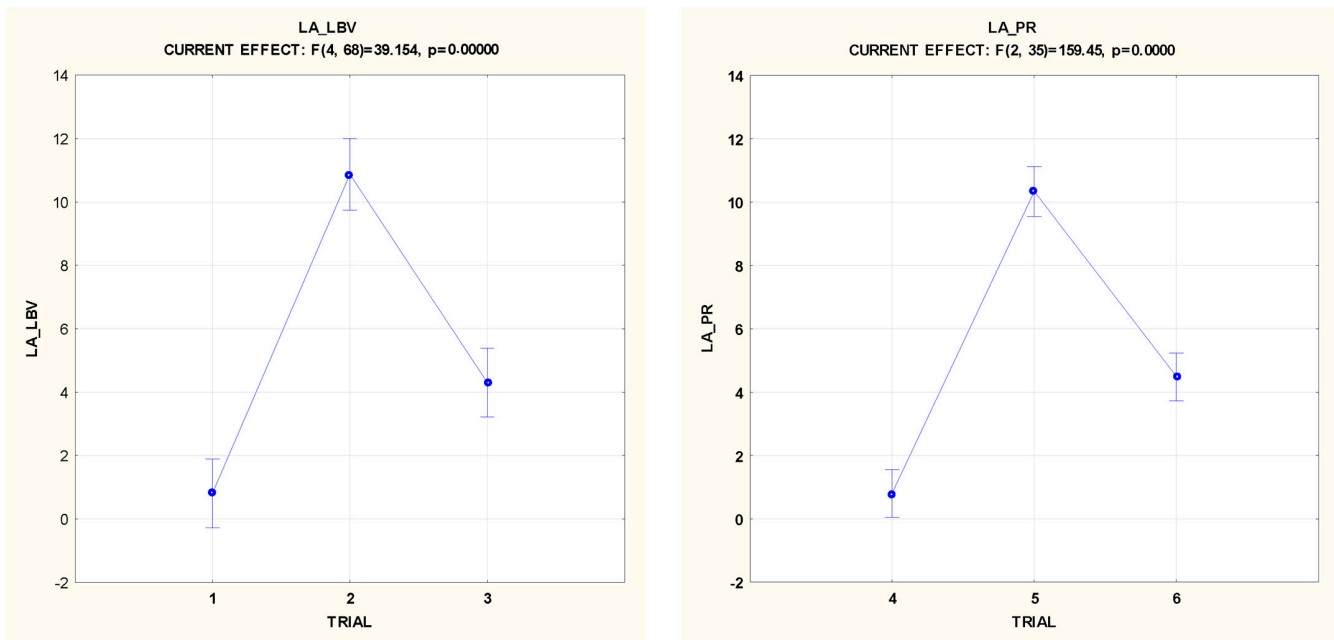

**Figure A6.** Lactate level measurements in blood plasma (mmol).

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
