# Peer review of "Assessment of the Effectiveness of Vibration Therapy and Passive Rest on the Recovery of Muscular Strength and Plasma Lactate Levels in the Upper Limbs after Intense Anaerobic Exercise in Elite Boxers and Kickboxers"

_applsci, doi:10.3390/app14177820_

Round 1

Reviewer 1 Report

Comments and Suggestions for Authors

The number of subjects is 18 in line 107, but 14 in line 131. How many subjects were ultimately tested? I think there may be differences between boxers who use their arms primarily and kickboxers who use their legs as well, so the authors need to explain why two groups were used. Also, since one subject participated in two experiments, the authors need to state how long the washout period was.

The most important point is that the authors seem to equate muscle fatigue with muscle damage. The exercise protocol used in this study was a muscle-fatiguing exercise that is not associated with muscle damage. Moreover, the lactic acid used in this study is an indicator of muscle fatigue, and LDH is used as an indicator of muscle damage. Please delete any descriptions of muscle damage or previous research results related to this topic in the introduction and focus on the recovery period after intense exercise when muscle fatigue occurs.

Blood samples for lactate measurement need to be described as either fingertip or antecubital vein..

The results figures look like they were copied directly from a statistical program. Please use Excel or something similar to make the figures clearer.

Since Table 2 is too long, I think it is unnecessary to describe the contents of the table that are already described in the figures.

Author Response

Dear Reviewer,

Thank you very much for your time and valuable comments, which all have been considered and incorporated. The detailed list of responses is given below. We hope that the modifications and explanation will be acceptable for you.

Yours sincerely,

Rydzik, corresponding author

The number of subjects is 18 in line 107, but 14 in line 131. How many subjects were ultimately tested? I think there may be differences between boxers who use their arms primarily and kickboxers who use their legs as well, so the authors need to explain why two groups were used. Also, since one subject participated in two experiments, the authors need to state how long the washout period was.

A: Although 18 elite boxers and kickboxers aged 19-32 years were initially recruited, only 14 completed all phases of the study due to personal reasons or scheduling conflicts

The most important point is that the authors seem to equate muscle fatigue with muscle damage. The exercise protocol used in this study was a muscle-fatiguing exercise that is not associated with muscle damage. Moreover, the lactic acid used in this study is an indicator of muscle fatigue, and LDH is used as an indicator of muscle damage. Please delete any descriptions of muscle damage or previous research results related to this topic in the introduction and focus on the recovery period after intense exercise when muscle fatigue occurs.

A: This has been corrected 

Blood samples for lactate measurement need to be described as either fingertip or antecubital vein..

A: This has been corrected "Blood samples for lactate measurement were taken from the fingertip and measured with a lactat scoute device"

The results figures look like they were copied directly from a statistical program. Please use Excel or something similar to make the figures clearer.

A: The charts are from the statistica program, we realize that they do not look attractive, so we decided to remove them , and put the information in the text. In addition, we left them in the supplementary material 

Since Table 2 is too long, I think it is unnecessary to describe the contents of the table that are already described in the figures.

A: Due to the removal of the figures, leaving the table is essential. We tried to expand the legend to make it easier to read 

Reviewer 2 Report

Comments and Suggestions for Authors

General remarks

Overall the quality of the manuscript is correct, but there are several points at which significant improvements, clarifications must be conducted

Specified remarks

„Maximum punching strength was measured based on three punches with each upper limb using accelerometric sensors” line 196. acceleration, strength, force are not interchangeable words, values. Must be clarified

„Maximum strength capabilities were measured under isokinetic contraction conditions for the flexors and extensors of the elbow joints at an angular velocity of ω=300 °s-1 using a Biodex System 4 Pro dynamometer” line 202. again, strength, torque, physical variables must be used accordingly.

„Participants performed auxotonic exercises over three rounds of alternating 180 straight punches in each round using a boxing trainer over 120 seconds at no less than 80% of their maximum punching force” line 210.  it is not clear how auxotonic contraction was controlled. Was punch force measured?

„Physical exertion and muscle fatigue were monitored using the accelerometric sensors to ensure participants reached a state of fatigue” line 213. it is unclear how this was conducted

How VT protocol duration, frequency was determined?

„The VT group achieved peak torque faster than the passive rest (PR)  group” line 238. What does faster mean. how was this determined? from which baselne?

„The acceleration time (T_ACC), measuring the speed at which muscles reach a set velocity, was significantly shorter in the VT group” line 245. Acceleration time is defined by time, not speed. There is confusion between the physical variables.

What is the unit used in figure 1 and 2?

Section 3.3. Muscle work and fatigue is hard to interpret based ont he used data collection methods

Section 3.4. Muscle strength balance. I couldn’t find any evidence in the text how antagonistic ratio was determined. This must be addressed

Table 2 is hard to interpret

„The results of lactate level measurements in the blood indicated similar, statistically insignificant average values” this is meaningless

How was work determined?

How mush time was used between the two different protocols?

Was there familiarization included to Biodex? At 300deg/s familiarization is an important factor.

Were all measured datasets normally distributed???

Author Response

Dear Reviewer,

Thank you very much for your time and valuable comments, which all have been considered and incorporated. The detailed list of responses is given below. We hope that the modifications and explanation will be acceptable for you.

Yours sincerely,

Rydzik, corresponding author

General remarks

Overall the quality of the manuscript is correct, but there are several points at which significant improvements, clarifications must be conducted

Specified remarks

„Maximum punching strength was measured based on three punches with each upper limb using accelerometric sensors” line 196. acceleration, strength, force are not interchangeable words, values. Must be clarified

A:  the concept has been clarified 

„Maximum strength capabilities were measured under isokinetic contraction conditions for the flexors and extensors of the elbow joints at an angular velocity of ω=300 °s-1 using a Biodex System 4 Pro dynamometer” line 202. again, strength, torque, physical variables must be used accordingly.

A:  the concept has been clarified 

„Participants performed auxotonic exercises over three rounds of alternating 180 straight punches in each round using a boxing trainer over 120 seconds at no less than 80% of their maximum punching force” line 210.  it is not clear how auxotonic contraction was controlled. Was punch force measured?

A:  the concept has been clarified 

„Physical exertion and muscle fatigue were monitored using the accelerometric sensors to ensure participants reached a state of fatigue” line 213. it is unclear how this was conducted

A:  the concept has been clarified 

How VT protocol duration, frequency was determined?

A:  The duration and frequency of the VT (Vibration Therapy) protocol were determined based on existing literature and prior studies that have established the effectiveness of vibration therapy in muscle recovery. Specifically, the frequency range of 20 to 50 Hz was selected because it has been shown to optimize muscle stimulation without causing additional fatigue. The 15-minute duration was chosen to provide sufficient time for the therapeutic effects to manifest while avoiding overstimulation of the muscles. These parameters were designed to maximize recovery efficiency, particularly in elite athletes like those participating in the study.

„The VT group achieved peak torque faster than the passive rest (PR)  group” line 238. What does faster mean. how was this determined? from which baselne?

A:The term "faster" in this context refers to the shorter time taken by the VT (Vibration Therapy) group to achieve peak torque during isokinetic testing compared to the Passive Rest (PR) group. This was determined by measuring the "time to peak torque" (T_PTQ), which indicates the duration from the start of muscle contraction to the point where maximum torque is generated. The baseline for this measurement was the initial isokinetic strength assessment conducted before the muscle fatigue was induced and before the recovery interventions were applied. This baseline allowed for a direct comparison of the time taken to reach peak torque between the two recovery methods.

„The acceleration time (T_ACC), measuring the speed at which muscles reach a set velocity, was significantly shorter in the VT group” line 245. Acceleration time is defined by time, not speed. There is confusion between the physical variables.

A:This has been corrected 

What is the unit used in figure 1 and 2?

A: This has been corrected

Section 3.3. Muscle work and fatigue is hard to interpret based ont he used data collection methods

A: Work was determined by integrating the force-displacement curve over the range of motion during isokinetic testing 

Section 3.4. Muscle strength balance. I couldn’t find any evidence in the text how antagonistic ratio was determined. This must be addressed

A: The agonist-to-antagonist ratio was calculated by comparing the peak torque of opposing muscle groups during isokinetic testing

Table 2 is hard to interpret

A: We added a legend under the table explaining all abbreviations, so as not to duplicate the results on the reviewer we removed the figures and included them in the supplementary material 

„The results of lactate level measurements in the blood indicated similar, statistically insignificant average values” this is meaningless

A: This has been corrected

How was work determined?

A: Work was determined by integrating the force-displacement curve over the range of motion during isokinetic testing. Specifically, the work done by the muscles was calculated as the product of the force exerted and the distance over which the force was applied during each contraction. This method provides a precise measurement of the total mechanical work performed by the muscles during the testing protocols, allowing for an accurate assessment of muscle performance and endurance in both the VT and PR groups.

How mush time was used between the two different protocols??

A: It was 7 days, added information in the text 

Was there familiarization included to Biodex? At 300deg/s familiarization is an important factor.

A: Yes added information in the text 

Were all measured datasets normally distributed???

A: Yes 

Round 2

Reviewer 2 Report

Comments and Suggestions for Authors

Thank you for the corrections, but:

Force and acceleration are not the same. You measure acceleration but talk about punch force. This is incorrect. Greater acceleration MIGHT be the result of greater force producing capabilities in the muscles, or MIGHT can be translated to greater force at a punch. This MUST be clarified

Table 2 is still difficult to digest

Agonistic-antagonistic ratio is meaningless if not being measured at the same joint angle, this must be addressed

You explain to me how work was determined but this should be included into the text

Author Response

Thank you for the corrections, but:

Force and acceleration are not the same. You measure acceleration but talk about punch force. This is incorrect. Greater acceleration MIGHT be the result of greater force producing capabilities in the muscles, or MIGHT can be translated to greater force at a punch. This MUST be clarified

A: This has been corrected

Table 2 is still difficult to digest

A: I understand, but I would like to point out that we have already removed all the engravings and it's difficult to do anything with the table. Its cumbersome appearance is due to the rough version of the manuscript. After it's formatted, it will definitely be more legible. I hope this will be acceptable to you.

Agonistic-antagonistic ratio is meaningless if not being measured at the same joint angle, this must be addressed

A: This has been corrected 

You explain to me how work was determined but this should be included into the text

A: The explanation has been added to the text.